# POLICY TRANSFER WITH STRATEGY OPTIMIZATION

**Wenhao Yu, C. Karen Liu, Greg Turk**
School of Interactive Computing
Georgia Institute of Technology
Atlanta, GA
wenhaoyu@gatech.edu, {karenliu,turk}@cc.gatech.edu

## ABSTRACT

Computer simulation provides an automatic and safe way for training robotic control policies to achieve complex tasks such as locomotion. However, a policy trained in simulation usually does not transfer directly to the real hardware due to the differences between the two environments. Transfer learning using domain randomization is a promising approach, but it usually assumes that the target environment is close to the distribution of the training environments, thus relying heavily on accurate system identification. In this paper, we present a different approach that leverages domain randomization for transferring control policies to unknown environments. The key idea that, instead of learning a single policy in the simulation, we simultaneously learn a family of policies that exhibit different behaviors. When tested in the target environment, we directly search for the best policy in the family based on the task performance, without the need to identify the dynamic parameters. We evaluate our method on five simulated robotic control problems with different discrepancies in the training and testing environment and demonstrate that our method can overcome larger modeling errors compared to training a robust policy or an adaptive policy.

## 1 INTRODUCTION

Recent developments in Deep Reinforcement Learning (DRL) have shown the potential to learn complex robotic controllers in an automatic way with minimal human intervention. However, due to the high sample complexity of DRL algorithms, directly training control policies on the hardware still remains largely impractical for agile tasks such as locomotion.

A promising direction to address this issue is to use the idea of transfer learning which learns a model in a source environment and transfers it to a target environment of interest. In the context of learning robotic control policies, we can consider the real world the target environment and the computer simulation the source environment. Learning in simulated environment provides a safe and efficient way to explore large variety of different situations that a real robot might encounter. However, due to the model discrepancy between physics simulation and the real-world environment, also known as the Reality Gap (Boeing & Bräunl, 2012; Koos et al., 2010), the trained policy usually fails in the target environment. Efforts have been made to analyze the cause of the Reality Gap (Neunert et al., 2017) and to develop more accurate computer simulation (Tan et al., 2018) to improve the ability of a policy when transferred it to real hardware. Orthogonal to improving the fidelity of the physics simulation, researchers have also attempted to cross the reality gap by training more capable policies that succeed in a large variety of simulated environments. Our method falls into the second category.

To develop a policy capable of performing in various environments with different governing dynamics, one can consider to train a *robust policy* or to train an *adaptive policy*. In both cases, the policy is trained in environments with randomized dynamics. A robust policy is trained under a range of dynamics without identifying the specific dynamic parameters. Such a policy can only perform well if the simulation is a good approximation of the real world dynamics. In addition, for more agile motor skills, robust policies may appear over-conservative due to the uncertainty in the training environments. On the other hand, when an adaptive policy is used, it learns to first identify, implicitly or explicitly, the dynamics of its environment, and then selects the best action according

to the identified dynamics. Being able to act differently according to the dynamics allows the adaptive policy to achieve higher performance on a larger range of dynamic systems. However, when the target dynamics is notably different from the training dynamics, it may still produce sub-optimal results for two reasons. First, when a sequence of novel observations is presented, the learned identification model in an adaptive policy may produce inaccurate estimations. Second, even when the identification model is perfect, the corresponding action may not be optimal for the new situation.

In this work, we introduce a new method that enjoys the versatility of an adaptive policy, while avoiding the challenges of system identification. Instead of relating the observations in the target environment to the similar experiences in the training environment, our method searches for the best policy directly based on the task performance in the target environment.

Our algorithm can be divided to two stages. The first stage trains a family of policies, each optimized for a particular vector of dynamic parameters. The family of policies can be parameterized by the dynamic parameters in a continuous representation. Each member of the family, referred to as a *strategy*, is a policy associated with particular dynamic parameters. Using a locomotion controller as an example, a strategy associated with low friction coefficient may exhibit cautious walking motion, while a strategy associated with high friction coefficient may result in more aggressive running motion. In the second stage we perform a search over the strategies in the target environment to find the one that achieves the highest task performance.

We evaluate our method on three examples that demonstrate transfer of a policy learned in one simulator DART, to another simulator MuJoCo. Due to the differences in the constraint solvers, these simulators can produce notably different simulation results. A more detailed description of the differences between DART and MuJoCo is provided in Appendix A. We also add latency to the MuJoCo environment to mimic a real world scenario, which further increases the difficulty of the transfer. In addition, we use a quadruped robot simulated in Bullet to demonstrate that our method can overcome actuator modeling errors. Latency and actuator modeling have been found to be important for Sim-to-Real transfer of locomotion policies (Tan et al., 2018; Neunert et al., 2017). Finally, we transfer a policy learned for a robot composed of rigid bodies to a robot whose end-effector is deformable, demonstrating the possiblity of using our method to transfer to problems that are challenging to model faithfully.

## 2 RELATED WORK

While DRL has demonstrated its ability to learn control policies for complex and dynamic motor skills in simulation (Schulman et al., 2015; 2017; Peng et al., 2018a; 2017; Yu et al., 2018; Heess et al., 2017), very few learning algorithms have successfully transferred these policies to the real world. Researchers have proposed to address this issue by optimizing or learning a simulation model using data from the real-world (Tan et al., 2018; 2016; Deisenroth & Rasmussen, 2011; Ha & Yamane, 2015; Abbeel & Ng, 2005). The main drawback for these methods is that for highly agile and high dimensional control problems, fitting an accurate dynamic model can be challenging and data inefficient.

Complementary to learning an accurate simulation model, a different line of research in sim-to-real transfer is to learn policies that can work under a large variety of simulated environments. One common approach is *domain randomization*. Training a robust policy with domain randomization has been shown to improve the ability to transfer a policy (Tan et al., 2018; Tobin et al., 2017; Rajeswaran et al., 2016; Pinto et al., 2017). Tobin et al. (2017) trained an object detector with randomized appearance and applied it in a real-world gripping task. Tan et al. (2018) showed that training a robust policy with randomized dynamic parameters is crucial for transferring quadruped locomotion to the real world. Designing the parameters and range of the domain to be randomized requires specific knowledge for different tasks. If the range is set too high, the policy may learn a conservative strategy or fail to learn the task, while a small range may not provide enough variation for the policy to transfer to real-world.

A similar idea is to train an adaptive policy with the current and the past observations as input. Such an adaptive policy is able to identify the dynamic parameters online either implicitly (OpenAI et al., 2018; Peng et al., 2018b) or explicitly (Yu et al., 2017) and apply actions appropriate for different system dynamics. Recently, adaptive policies have been used for sim-to-real transfer, such

as in-hand manipulation tasks (OpenAI et al., 2018) or non-prehensile manipulation tasks (Peng et al., 2018b). Instead of training one robust or adaptive policy, Zhang et al. (2018) trained multiple policies for a set of randomized environments and learned to combine them linearly in a separate set of environments. The main advantage of these methods is that they can be trained entirely in simulation and deployed in real-world without further fine-tuning. However, policies trained in simulation may not generalize well when the discrepancy between the target environment and the simulation is too large. Our method also uses dynamic randomization to train policies that exhibit different strategies for different dynamics, however, instead of relying on the simulation to learn an identification model for selecting the strategy, we propose to directly optimize the strategy in the target environment.

A few recent works have also proposed the idea of training policies in a source environment and fine-tune it in the target environment. For example, Cully et al. (2015) proposed MAP-Elite to learn a large set of controllers and applied Bayesian optimization for fast adaptation to hardware damages. Their approach searches for individual controllers for discrete points in a behavior space, instead of a parameterized family of policies as in our case, making it potentially challenging to be applied to higher dimensional behavior spaces. Rusu et al. (2016) used progressive network to adapt the policy to new environments by designing a policy architecture that can effectively utilize previously learned representations. Chen et al. (2018) learned an implicit representation of the environment variations by optimizing a latent policy input for each discrete instance of the environment. They showed that fine-tuning on this learned policy achieved improved learning efficiency. In contrast to prior work in which the fine-tuning phase adjusts the neural network weights in the target environment, we optimize only the dynamics parameters input to the policy. This allows our policies to adapt to the target environments with less data and to use sparse reward signal.

## 3 BACKGROUND

We formulate the motor skill learning problem as a Markov Decision Process (MDP), $\mathcal{M} = (\mathcal{S}, \mathcal{A}, r, \mathcal{P}, p_0, \gamma)$, where $\mathcal{S}$ is the state space, $\mathcal{A}$ is the action space, $r : \mathcal{S} \times \mathcal{A} \mapsto \mathbb{R}$ is the reward function, $\mathcal{P} : \mathcal{S} \times \mathcal{A} \mapsto \mathcal{S}$ is the transition function, $p_0$ is the initial state distribution and $\gamma$ is the discount factor. The goal of reinforcement learning is to find a control policy $\pi : \mathcal{S} \mapsto \mathcal{A}$ that maximizes the expected accumulated reward: $J_{\mathcal{M}}(\pi) = \mathbb{E}_{\tau=(s_0, a_0, ..., s_T)} \sum_{t=0}^{T} \gamma^t r(s_t, a_t)$, where $s_0 \sim p_0$, $a_t \sim \pi(s_t)$ and $s_{t+1} = \mathcal{P}(s_t, a_t)$. In practice, we usually only have access to an observation of the robot that contains a partial information of the robot's state. In this case, we will have a Partially-Observable Markov Decision Process (POMDP) and the policy would become $\pi : \mathcal{O} \mapsto \mathcal{A}$, where $\mathcal{O}$ is the observation space.

In the context of transfer learning, we can define a source MDP $\mathcal{M}^s$ and a target MDP $\mathcal{M}^t$ and the goal would be to learn a policy $\pi^s$ for $\mathcal{M}^s$ such that it also works well on $\mathcal{M}^t$. In this work, $\mathcal{P}$ is regarded as a parameterized space of transition functions, $s_{t+1} = \mathcal{P}_\mu(s_t, a_t)$, where $\mu$ is a vector of physical parameters defining the dynamic model (e.g. friction coefficient). The transfer learning in this context learns a policy under $\mathcal{P}^s$ and transfers to $\mathcal{P}^t$, where $\mathcal{P}^s \neq \mathcal{P}^t$.

## 4 METHODS

We propose a new method for transferring a policy learned in simulated environment to a target environment with unknown dynamics. Our algorithm consists of two stages: learning a family of policies and optimizing strategy.

### 4.1 LEARNING A FAMILY OF POLICIES

The first stage of our method is to learn a family of policies, each for a particular dynamics $\mathcal{P}_\mu^s(\cdot)$. One can potentially train each policy individually and interpolate them to cover the space of $\mu$ (Stulp et al., 2013; Da Silva et al., 2012). However, as the dimension of $\mu$ increases, the number of policies required for interpolation grows exponentially.

Since many of these policies are trained under similar dynamics, our method merges them into one neural network and trains the entire family of policies simultaneously. We follow the work by Yu

et al. (2017), which trains a policy $\pi : (\boldsymbol{o}, \mu) \mapsto \boldsymbol{a}$ that takes as input not only the observation of the robot $\boldsymbol{o}$, but also the physical parameters $\mu$. At the beginning of each rollout during the training, we randomly pick a new set of physical parameters for the simulation and fix it throughout the rollout. After training the policy this way, we obtain a family of policies that is parameterized by the dynamics parameters $\mu$. Given a particular $\mu$, we define the corresponding policy as $\pi_\mu : \boldsymbol{o} \mapsto \boldsymbol{a}$. We will call such an instantiated policy a *strategy*.

## 4.2 OPTIMIZATING STRATEGY

The second stage of our method is to search for the optimal strategy in the space of $\mu$ for the target environment. Previous work learns a mapping between the experiences under source dynamics $\mathcal{P}^s_\mu$ and the corresponding $\mu$. When new experiences are generated in the target environment, this mapping will identify a $\mu$ based on similar experiences previously generated in the source environment. While using experience similarity as a metric to identify $\mu$ transfers well to a target environment that has the same dynamic parameter space (Yu et al., 2017), it does not generalize well when the dynamic parameter space is different.

Since our goal is to find a strategy that works well in the target environment, a more direct approach is to use the performance of the task, i.e. the accumulated reward, in the target environment as the metric to search for the strategy:

$$\mu^* = \arg\max_\mu J_{\mathcal{M}^t}(\pi_\mu). \tag{1}$$

Solving Equation 1 can be done efficiently because the search space in Equation 1 is the space of dynamic parameters $\mu$, rather than the space of policies, which are represented as neural networks in our implementation. To further reduce the number of samples from the target environment needed for solving Equation 1, we investigated a number of algorithms, including Bayesian optimization, model-based methods and an evolutionary algorithm (CMA). A detailed description and comparison of these methods are provided in Appendix C.

We chose Covariance Matrix Adaptation (CMA) (Hansen et al., 1995), because it reliably outperforms other methods in terms of sample-efficiency. At each iteration of CMA, a set of samples are drawn from a Gaussian distribution over the space of $\mu$. For each sample, we instantiate a strategy $\pi_\mu$ and use it to generate rollouts in the target environment. The fitness of the sample is determined by evaluating the rollouts using $J_{\mathcal{M}^t}$. Based on the fitness values of the samples in the current iteration, the mean and the covariance matrix of the Gaussian distribution are updated for the next iteration.

## 5 EXPERIMENTS

To evaluate the ability of our method to overcome the reality gap, we train policies for four locomotion control tasks (hopper, walker2d, half cheetah, quadruped robot) and transfer each policy to environments with different dynamics. To mimic the reality gap seen in the real-world, we use target environments that are different from the source environments in their contact modeling, latency or actuator modeling. In addition, we also test the ability of our method to generalize to discrepancies in body mass, terrain slope and end-effector materials. Figure 1 shows the source and target environments for all the tasks and summarizes the modeled reality gap in each task. During training, we choose different combinations of dynamic parameters to randomize and make sure they do not overlap with the variations in the testing environments. For clarity of exposition, we denote the dimension of the dynamic parameters that are randomized during training as $dim(\mu)$. For all examples, we use the Proximal Policy Optimization (PPO) (Schulman et al., 2017) to optimize the control policy. A more detailed description of the experiment setup as well as the simulated reality gaps are provided in Appendix B. For each example presented, we run three trials with different random seeds and report the mean and one standard deviation for the total reward.

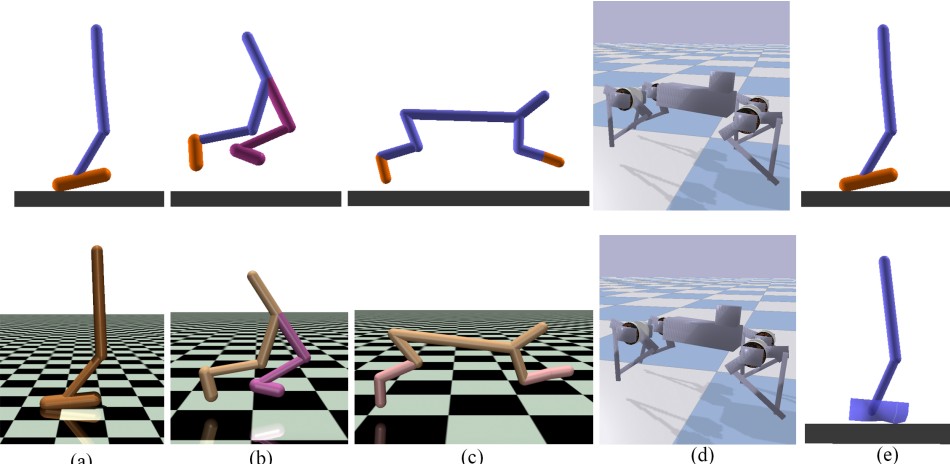

Figure 1: The environments used in our experiments. Environments in the top row are source environments and environments in the bottom row are the target environments we want to transfer the policy to. (a) Hopper from DART to MuJoCo. (b) Walker2d from DART to MuJoCo with latency. (c) HalfCheetah from DART to MuJoCo with latency. (d) Minitaur robot from inaccurate motor modeling to accurate motor modeling. (e) Hopper from rigid to soft foot.

## 5.1 BASELINE METHODS

We compare our method, Strategy Optimization with CMA-ES (SO-CMA) to three baseline methods: training a robust policy (Robust), training an adaptive policy (Hist) and training a Universal Policy with Online System Identification (UPOSI) (Yu et al., 2017). The robust policy is represented as a feed forward neural network, which takes as input the most recent observation from the robot, i.e. $\pi_{robust} : \boldsymbol{o} \mapsto \boldsymbol{a}$. The policy needs to learn actions that work for all the training environments, but the dynamic parameters cannot be identified from its input. In contrast, an adaptive policy is given a history of observations as input, i.e. $\pi_{adapt} : (\boldsymbol{o}_{t-h}, \ldots, \boldsymbol{o}_t) \mapsto \boldsymbol{a}_t$. This allows the policy to potentially identify the environment being tested and adaptively choose the actions based on the identified environment. There are many possible ways to train an adaptive policy, for example, one can use an LSTM network to represent the policy or use a history of observations as input to a feed-forward network. We find that for the tasks we demonstrate, directly training an LSTM policy using PPO is much less efficient and reaches lower end performance than training a feed-forward network with history input. Therefore, in our experiments we use a feed-forward network with a history of 10 observations to represent the adaptive policy $\pi_{adapt}$. We also compare our method to UPOSI, which decouples the learning of an adaptive policy into training a universal policy via reinforcement learning and a system identification model via supervised learning. In theory UPOSI and Hist should achieve similar performance, while in practice we expect UPOSI to learn more efficiently due to the decoupling. We adopt the same training procedure as done by Yu et al. (2017), and use a history of 10 observations as input to the online system identification model. For fair comparison, we continue to train the baseline methods after transferring to the target environment, using the same amount of samples SO-CMA consumes in the target environment. We refer this additional training step as 'fine-tuning'. In addition to the baseline methods, we also compare our method to the performance of policies trained directly in the target environments, which serves as an 'Oracle' benchmark. The Oracle policies for Hopper, Walke2d, HalfCheetah and Hopper Soft was trained for $1,000,000$ samples in the target environment as in Schulman et al. (2017). For the quadruped example, we run PPO for $5,000,000$ samples, similar to Tan et al. (2018). We detail the process of 'fine-tuning' in Appendix B.4

## 5.2 HOPPER DART TO MUJOCO

In the first example, we build a single-legged robot in DART similar to the Hopper environment simulated by MuJoCo in OpenAI Gym (Brockman et al., 2016). We investigate two questions in this example: 1) does SO-CMA work better than alternative methods in transferring to unknown environments? and 2) how does the choice of $dim(\mu)$ affect the performance of policy transfer? To

this end, we perform experiments with $dim(\mu) = 2, 5$ and $10$. For the experiment with $dim(\mu) = 2$, we randomize the mass of the robot's foot and the restitution coefficient between the foot and the ground. For $dim(\mu) = 5$, we in addition randomize the friction coefficient, the mass of the robot's torso and the joint strength of the robot. We further include the mass of the rest two body parts and the joint damping to construct the randomized dynamic parameters for $dim(\mu) = 10$. The specific ranges of randomization are described in Appendix B.4.

We first evaluate how the performance of different methods varies with the number of samples in the target environment. As shown in Figure 2, when $dim(\mu)$ is low, none of the four methods were able to transfer to the MuJoCo Hopper successfully. This is possibly due to there not being enough variation in the dynamics to learn diverse strategies. When $dim(\mu) = 5$, SO-CMA can successfully transfer the policy to MuJoCo Hopper with good performance, while the baseline methods were not able to adapt to the new environment using the same sample budget. We further increase $dim(\mu)$ to 10 as shown in Figure 2 (c) and find that SO-CMA achieved similar end performance to $dim(\mu) = 5$, while the baselines do not transfer well to the target environment.

We further investigate whether SO-CMA can generalize to differences in joint limits in addition to the discrepancies between DART and MuJoCo. Specifically, we vary the magnitude of the ankle joint limit in $[0.5, 1.0]$ radians (default is $0.785$) for the MuJoCo Hopper, and run all the methods with $30,000$ samples. The result can be found in Figure 3. We can see a similar trend that with low $dim(\mu)$ the transfer is challenging, and with higher value of $dim(\mu)$ SO-CMA is able to achieve notably better transfer performance than the baseline methods.

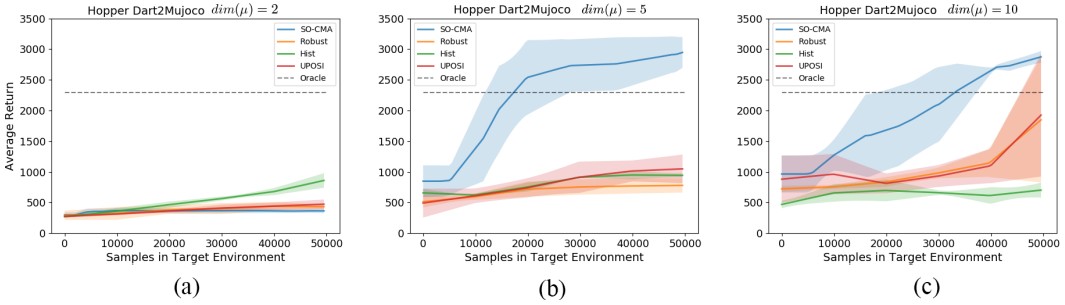

Figure 2: Transfer performance vs Sample number in target environment for the Hopper example. Policies are trained to transfer from DART to MuJoCo.

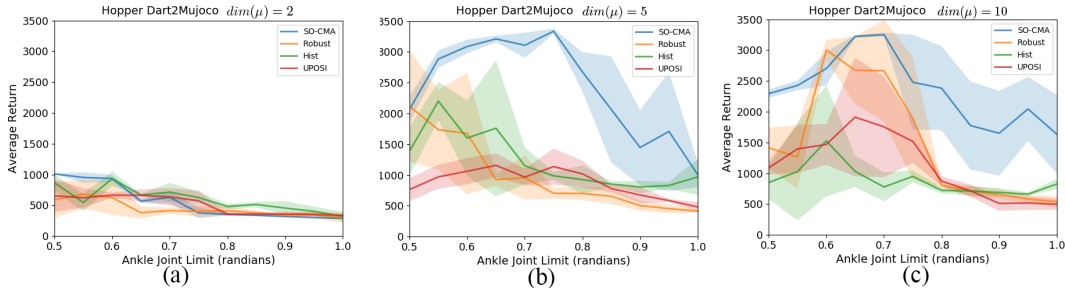

Figure 3: Transfer performance for the Hopper example. Policies are traiend to transfer from DART to MuJoCo with different ankle joint limits (horizontal axis). All trials run with total sample number of $30,000$ in the target environment.

## 5.3 WALKER2D DART TO MUJOCO WITH LATENCY

In this example, we use the lower body of a biped robot constrained to a $2D$ plane, according to the Walker2d environment in OpenAI Gym. We find that with different initializations of the policy network, training could lead to drastically different gaits, e.g. hopping with both legs, running with one legs dragging the other, normal running, etc. Some of these gaits are more robust to environment changes than others, which makes analyzing the performance of transfer learning algorithms

challenging. To make sure the policies are more comparable, we use the symmetry loss from Yu et al. (2018), which leads to all policies learning a symmetric running gait. To mimic modeling error seen on real robots, we add a latency of 8ms to the MuJoCo simulator. We train policies with $dim(\mu) = 8$, for which we randomize the friction coefficient, restitution coefficient and the joint damping of the six joints during training. Figure 4 (a) shows the transfer performance of different method with respect to the sample numbers in the target environment.

We further vary the mass of the robot's right foot in $[2, 9]$kg in the MuJoCo Walker2d environment and compare the transfer performance of SO-CMA to the baselines. The default foot mass is $2.9$ kg. We use in total $30,000$ samples in the target environment for all methods being compared and the results can be found in Figure 4 (b). In both cases, our method achieves notably better performance than Hist and UPOSI, while being comparable to Robust.

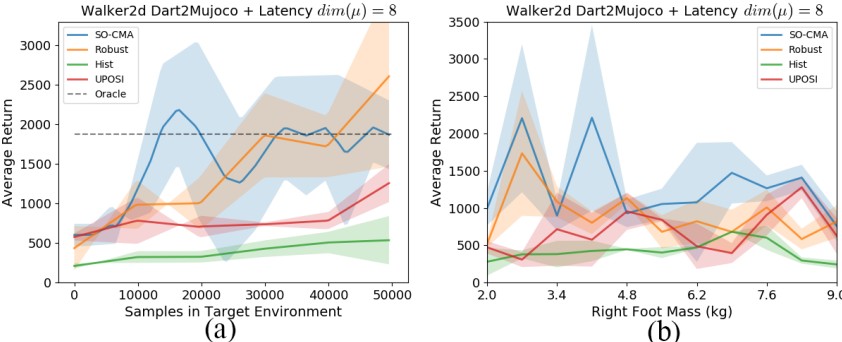

Figure 4: Transfer performance for the Walker2d example. (a) Transfer performance vs sample number in target environment on flat surface. (b) Transfer performance vs foot mass, trained with $30,000$ samples in the target environment.

## 5.4 HALFCHEETAH DART TO MUJOCO WITH DELAY

In the third example, we train policies for the HalfCheetah environment from OpenAI Gym. We again test the performance of transfer from DART to MuJoCo for this example. In addition, we add a latency of 50ms to the target environment. We randomize 11 dynamic parameters in the source environment consisting of the mass of all body parts, the friction coefficient and the restitution coefficient during training, i.e. $dim(\mu) = 11$. The results of the performance with respect to sample numbers in target environment can be found in Figure 5 (a). We in addition evaluate transfer to environments where the slope of the ground varies, as shown in Figure 5 (b). We can see that SO-CMA outperforms Robust and Hist, while achieves similar performance as UPOSI.

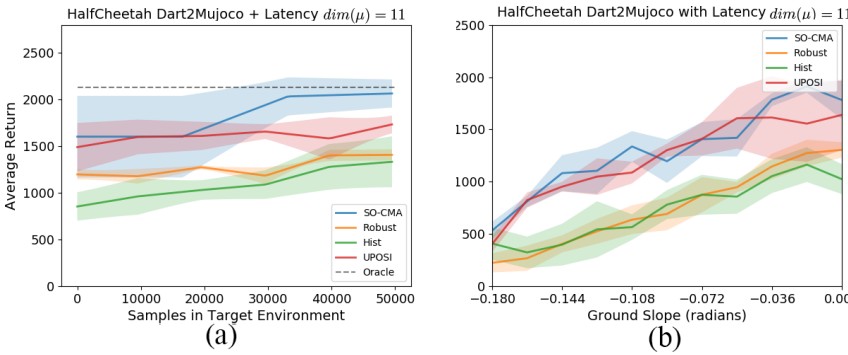

Figure 5: Transfer performance for the HalfCheetah example. (a) Transfer performance vs sample number in target environment on flat surface. (b) Transfer performance vs surface slope, trained with $30,000$ samples in the target environment.

## 5.5 QUADRUPED ROBOT WITH ACTUATOR MODELING ERROR

As demonstrated by Tan et al. (2018), when a robust policy is used, having an accurate actuator model is important to the successful transfer of policy from simulation to real-world for a quadruped robot, Minitaur (Figure 1 (d)). Specifically, they found that when a linear torque-current relation is assumed in the actuator dynamics in the simulation, the policy learned in simulation transfers poorly to the real hardware. When the actuator dynamics is modeled more accurately, in their case using a non-linear torque-current relation, the transfer performance were notably improved.

In our experiment, we investigate whether SO-CMA is able to overcome the error in actuator models. We use the same simulation environment from Tan et al. (2018), which is simulated in Bullet (Coumans & Bai, 2016-2017). During the training of the policy, we use a linear torque-current relation for the actuator model, and we transfer the learned policy to an environment with the more accurate non-linear torque-current relation. We use the same 25 dynamic parameters and corresponding ranges used by Tan et al. (2018) for dynamics randomization during training. When applying the robust policy to the accurate actuator model, we observe that the quadruped tends to sink to the ground, similar to what was observed by Tan et al. (2018). SO-CMA, on the other hand, can successfully transfer a policy trained with a crude actuator model to an environment with more realistic actuators(Figure 6 (a)).

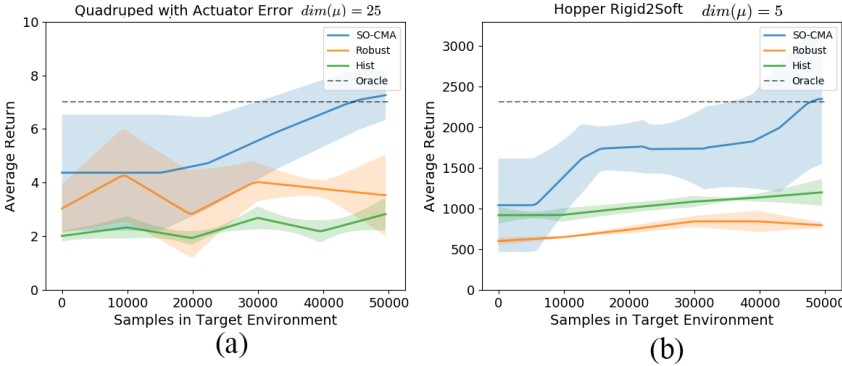

Figure 6: Transfer performance for the Quadruped example (a) and the Soft-foot Hopper example (b).

## 5.6 HOPPER RIGID TO DEFORMABLE FOOT

Applying deep reinforcement learning to environments with deformable objects can be computationally inefficient (Clegg et al., 2018). Being able to transfer a policy trained in a purely rigid-body environment to an environment containing deformable objects can greatly improve the efficiency of learning. In our last example, we transfer a policy trained for the Hopper example with rigid objects only to a Hopper model with a deformable foot (Figre 1 (e)). The soft foot is modeled using the soft shape in DART, which uses an approximate but relatively efficient way of modeling deformable objects (Jain & Liu, 2011). We train policies in the rigid Hopper environment and randomize the same set of dynamic parameters as in the in the DART-to-MuJoCo transfer example with $\dim(\mu) = 5$. We then transfer the learned policy to the soft Hopper environment where the Hopper's foot is deformable. The results can be found in Figure 6 (b). SO-CMA is able to successfully control the robot to move forward without falling, while the baseline methods fail to do so.

## 6 DISCUSSIONS

We have demonstrated that our method, SO-CMA, can successfully transfer policies trained in one environment to a notably different one with a relatively low amount of samples. One advantage of SO-CMA, compared to the baselines, is that it works consistently well across different examples, while none of the baseline methods achieve successful transfer for all the examples.

We hypothesize that the large variance in the performance of the baseline methods is due to their sensitivity to the type of task being tested. For example, if there exists a robust controller that

works for a large range of different dynamic parameters $\mu$ in the task, such as a bipedal running motion in the Walker2d example, training a Robust policy may achieve good performance in transfer. However, when the optimal controller is more sensitive to $\mu$, Robust policies may learn to use overly-conservative strategies, leading to sub-optimal performance (e.g. in HalfCheetah) or fail to perform the task (e.g. in Hopper). On the other hand, if the target environment is not significantly different from the training environments, UPOSI may achieve good performance, as in HalfCheetah. However, as the reality gap becomes larger, the system identification model in UPOSI may fail to produce good estimates and result in non-optimal actions. Furthermore, Hist did not achieve successful transfer in any of the examples, possibly due to two reasons: 1) it shares similar limitation to UPOSI when the reality gap is large and 2) it is in general more difficult to train Hist due to the larger input space, so that with a limited sample budget it is challenging to fine-tune Hist effectively.

We also note that although in some examples certain baseline method may achieve successful transfer, the fine-tuning process of these methods relies on having a dense reward signal. In practice, one may only have access to a sparse reward signal in the target environment, e.g. distance traveled before falling to the ground. Our method, using an evolutionary algorithm (CMA), naturally handles sparse rewards and thus the performance gap between our method (SO-CMA) and the baseline methods will likely be large if a sparse reward is used.

## 7 CONCLUSION

We have proposed a policy transfer algorithm where we first learn a family of policies simultaneously in a source environment that exhibits different behaviors and then search directly for a policy in the family that performs the best in the target environment. We show that our proposed method can overcome large modeling errors, including those commonly seen on real robotic platforms with relatively low amount of samples in the target environment. These results suggest that our method has the potential to transfer policies trained in simulation to real hardware.

There are a few interesting directions that merit further investigations. First, it would be interesting to explore other approaches for learning a family of policies that exhibit different behaviors. One such example is the method proposed by Eysenbach et al. (2018), where an agent learns diverse skills without a reward function in an unsupervised manner. Another example is the HCP-I policy proposed by Chen et al. (2018), which learns a latent representation of the environment variations implicitly. Equipping our policy with memories is another interesting direction to investigate. The addition of memory will extend our method to target environments that vary over time. We have investigated in a few options for strategy optimization and found that CMA-ES works well for our examples. However, it would be desired if we can find a way to further reduce the sample required in the target environment. One possible direction is to warm-start the optimization using models learned in simulation, such as the calibration model in Zhang et al. (2018) or the online system identification model in Yu et al. (2017).

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

## A    Differences between DART and MuJoCo

DART (Lee et al., 2018) and MuJoCo (Todorov et al., 2012) are both physically-based simulators that computes how the state of virtual character or robot evolves over time and interacts with other objects in a physical way. Both of them have been demonstrated for transferring controllers learned for a simulated robot to a real hardware (Tan et al., 2018; 2016), and there has been work trying to transfer policies between DART and MuJoCo (Wulfmeier et al., 2017). The two simulators are similar in many aspects, for example both of them uses generalized coordinates for representing the state of a robot. Despite the many similarities between DART and MuJoCo, there are a few important differences between them that makes transferring a policy trained in one simulator to the other challenging. For the examples of DART-to-MuJoCo transfer presented in this paper, there are three major differences as described below:

1. Contact Handling

   Contact modeling is important for robotic control applications, especially for locomotion tasks, where robots heavily rely on manipulating contacts between end-effector and the ground to move forward. In DART, contacts are handled by solving a linear complementarity problem (LCP) (Tan et al.), which ensures that in the next timestep, the objects will not penetrate with each other, while satisfying the laws of physics. In MuJoCo, the contact dynamics is modeled using a complementarity-free formulation, which means the objects might penetrate with each other. The resulting impulse will increase with the penetration depth and separate the penetrating objects eventually.

2. Joint Limits

   Similar to the contact solver, DART tries to solve the joint limit constraints exactly so that the joint limit is not violated in the next timestep, while MuJoCo uses a soft constraint formulation, which means the character may violate the joint limit constraint.

3. Armature

   In MuJoCo, a diagonal matrix $\sigma \mathbb{I}_n$ is added to the joint space inertia matrix that can help stabilize the simulation, where $\sigma \in \mathbb{R}$ is a scalar named Armature in MuJoCo and $\mathbb{I}_n$ is the $n \times n$ identity matrix. This is not modeled in DART.

To illustrate how much difference these simulator characteristics can lead to, we compare the Hopper example in DART and MuJoCo by simulating both using the same sequence of randomly generated actions from an identical state. We plot the linear position and velocity of the torso and foot of the robot, which is shown in Figure 7. We can see that due to the differences in the dynamics, the two simulators would control the robot to reach notably different states even though the initial state and control signals are identical.

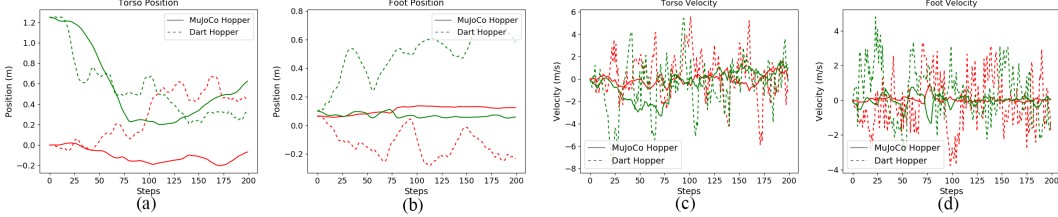

Figure 7: Comparison of DART and MuJoCo environments under the same control signals. The red curves represent position or velocity in the forward direction and the green curves represent position or velocity in the upward direction.

## B    Experiment Details

### B.1    Experiment Settings

We use Proximal Policy Optimization (PPO) implemented in OpenAI Baselines (Dhariwal et al., 2017) for training all the policies in our experiments. For simulation in DART, we use DartEnv (Yu

Table 1: Environment Details

| Environment | Observation | Action | Reward |
|---|---|---|---|
| Hopper | 11 | 3 | $s_t^{vel} - 0.001||a_t||_2^2 + 1$ |
| Walker2d | 17 | 6 | $s_t^{vel} - 0.001||a_t||_2^2 + 1$ |
| HalfCheetah | 17 | 6 | $s_t^{vel} - 0.1||a_t||_2^2 + 1$ |
| Quadruped | 12 | 8 | $s_t^{vel}\Delta t - 0.008\Delta t|a_t \cdot \dot{q}_t|$ |

& Liu, 2017), which implements the continuous control benchmarks in OpenAI Gym using PyDart (Ha, 2016). For all of our examples, we represent the policy as a feed-forward neural network with three hidden layers, each consists of $64$ hidden nodes.

## B.2 ENVIRONMENT DETAILS

The observation space, action space and the reward function used in all of our examples can be found in Table 1. For the Walker2d environment, we found that with the original environment settings in OpenAI Gym, the robot sometimes learn to hop forward, possibly due to the ankle being too strong. Therefore, we reduce the torque limit of the ankle joint in both DART and MuJoCo environment for the Walker2d problem from $[-100, 100]$ to $[-20, 20]$. We found that with this modification, we can reliably learn locomotion gaits that are closer to a human running gait.

Below we list the dynamic randomization settings used in our experiments. Table 2, Table 3 and Table 4 shows the range of the randomization for different dynamic parameters in different environments. For the quadruped example, we used the same settings as in Tan et al. (2018).

Table 2: Dynamic Randomization details for Hopper

| Dynamic Parmeter | Range |
|---|---|
| Friction Coefficient | $[0.2, 1.0]$ |
| Restitution Coefficient | $[0.0, 0.3]$ |
| Mass | $[2.0, 15.0]$kg |
| Joint Damping | $[0.5, 3]$ |
| Joint Torque Scale | $[50\%, 150\%]$ |

Table 3: Dynamic Randomization details for Walker2d

| Dynamic Parmeter | Range |
|---|---|
| Friction Coefficient | $[0.2, 1.0]$ |
| Restitution Coefficient | $[0.0, 0.8]$ |
| Joint Damping | $[0.1, 3.0]$ |

## B.3 SIMULATED REALITY GAPS

To evaluate the ability of our method to overcome the modeling error, we designed six types of modeling errors. Each example shown in our experiments contains one or more modeling errors listed below.

1. DART to MuJoCo

   For the Hopper, Walker2d and HalfCheetah example, we trained policies that transfers from DART environment to MuJoCo environment. As discussed in Appendix A, the major differences between DART and MuJoCo are contacts, joint limits and armature.

2. Latency

   The second type of modeling error we tested is latency in the signals. Specifically, we model the latency between when an observation $o$ is sent out from the robot, and when the action corresponding to this observation $a = \pi(o)$ is executed on the robot. When a policy

Table 4: Dynamic Randomization details for HalfCheetah

| Dynamic Parmeter | Range |
|---|---|
| Friction Coefficient | $[0.2, 1.0]$ |
| Restitution Coefficient | $[0.0, 0.5]$ |
| Mass | $[1.0, 15.0]$kg |
| Joint Torque Scale | $[30\%, 150\%]$ |

is trained without any delay, it is usually very challenging to transfer it to problems with delay added. The value of delay is usually below 50ms and we use 8ms and 50ms in our examples.

3. Actuator Modeling Error

   As noted by Tan et al. (2018), error in actuator modeling is an important factor that contributes to the reality gap. They solved it by identifying a more accurate actuator model by fitting a piece-wise linear function for the torque-current relation. We use their identified actuator model as the ground-truth target environment in our experiments and used the ideal linear torque-current relation in the source environments.

4. Foot Mass

   In the example of Walker2d, we vary the mass of the right foot on the robot to create a family of target environments for testing. The range of the torso mass varies in $[2, 9]$kg.

5. Terrain Slope

   In the example of HalfCheetah, we vary the slope of the ground to create a family of target environments for testing. This is implemented as rotating the gravity direction by the same angle. The angle varies in the range $[-0.18, 0.0]$ radians.

6. Rigid to Deformable

   The last type of modeling error we test is that a deformable object in the target environment is modeled as a rigid object in the source environment. The deformable object is modeled using the soft shape object in DART. In our example, we created a deformable box of size $0.5m \times 0.19m \times 0.13m$ around the foot of the Hopper. We set the stiffness of the deformable object to be $10,000$ and the damping to be $1.0$. We refer readers to Jain & Liu (2011) for more details of the softbody simulation.

## B.4   POLICY TRAINING

For training policies in the source environment, we run PPO for 500 iterations. In each iteration, we sample $40,000$ steps from the source environment to update the policy. For the rest of the hyperparameters, we use the default value from OpenAI Baselines (Dhariwal et al., 2017). We use a large batch size in our experiments as the policy needs to be trained to work on different dynamic parameters $\mu$.

For fine-tuning of the Robust and Adaptive policy in the target environment, we sample $2,000$ steps from the target environment at each iteration of PPO, which is the default value used in OpenAI Baselines. Here we use a smaller batch size for two reasons: 1) since the policy is trained to work on only one dynamics, we do not need as many samples to optimize the policy in general and 2) the fine-tuning process has a limited sample budget and thus we want to use a smaller batch size so that the policy can be improved more. In the case where we use a maximum of $50,000$ samples for fine-tuning, this amounts to 50 iterations of PPO updates. Furthermore, we use a maximum rollout length of $1,000$, while the actual length of the rollout collected during training is general shorter due to the early termination, e.g. when the robot falls to the ground. Therefore, with $50,000$ samples in total, the fine-tuning process usually consists of $100 \sim 300$ rollouts, depending on the task.

## B.5   STRATEGY OPTIMIZATION WITH CMA-ES

We use the CMA-ES implementation in python by (PyC). At each iteration of CMA-ES, we generate $4 + \lfloor 3 * log(N) \rfloor$ samples from the latest Gaussian distribution, where $N$ is the dimension of the

dynamic parameters. During evaluation of each sample $\mu_i$, we run the policy $\pi_{\mu_i}$ in the target environment for three trials and average the returns to obtain the fitness of this sample.

## C    ALTERNATIVE METHODS FOR STRATEGY OPTIMIZATION

In addition to CMA-ES, we have also experimented with a few other options for finding the best $\mu$ such that $\pi_\mu$ works well in the target environment. Here we show some experiment results for Strategy Optimization with Bayesian Optimization (SO-BO) and Model-based Optimization (SO-MB).

### C.1    BAYESIAN OPTIMIZATION

Bayesian Optimization is a gradient-free optimization method that is known to work well for low dimensional continuous problems where evaluating the quality of each sample can be expensive. The main idea in Bayesian optimization is to incrementally build a Gaussian process (GP) model that estimates the loss of a given search parameter. At each iteration, a new sample is drawn by optimizing an acquisition function on the GP model. The acquisition function takes into account the exploration (search where the GP has low uncertainty) and exploitation (search where the GP predicts low loss). The new sample is then evaluated and added to the training dataset for GP.

We test Bayesian Optimization on the Hopper and Quadruped example, as shown in Figure 8. We can see that Bayesian Optimization can achieve comparable performance as CMA-ES and thus is a viable choice to our problem. However, SO-BA appears in general noisier than CMA-ES and is in general less computationally efficient due to the re-fitting of GP models.

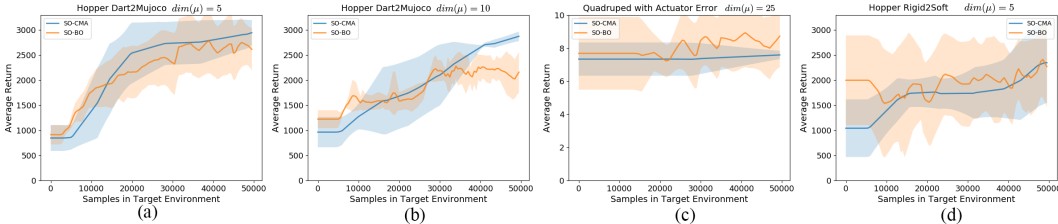

Figure 8: Comparison of SO-CMA and SO-BA for Hopper and Quadruped examples.

### C.2    MODEL-BASED OPTIMIZATION

Another possible way to perform strategy optimization is to use a model-based method. In a model-based method, we learn the dynamics of the target environment using generic models such as neural networks, Gaussian process, linear functions, etc. After we have learned a dynamics model, we can use it as an approximation of the target environment to optimize $\mu$.

We first tried using feed-forward neural networks to learn the dynamics and optimize $\mu$. However, this method was not able to reliably find $\mu$ that lead to good performance. This is possibly due to that any error in the prediction of the states would quickly accumulate over time and lead to inaccurate predictions. In addition, this method would not be able to handle problems where latency is involved.

In the experiments presented here, we learn the dynamics of the target environment with a Long Short Term Memory (LSTM) network (Hochreiter & Schmidhuber, 1997). Given a target environment, we first sample $\mu$ uniformly and collect experience using $\pi_\mu$ until we have 5,000 samples. We use these samples to fit an initial LSTM dynamic model. We then alternate between finding the best dynamic parameters $\hat{\mu}$ such that $\pi_{\hat{\mu}}$ achieves the best performance under the latest LSTM dynamic model and update the LSTM dynamic model using data generated from $\pi_{\hat{\mu}}$. This is repeated until we have reached the sample budget.

We found that LSTM notably outperformed feed-forward networks when applied to strategy optimization. One result for Hopper DART-to-MuJoCo can be found in Figure 9. It can be seen that Model-based method with LSTM is able to achieve similar performance as CMA-ES.

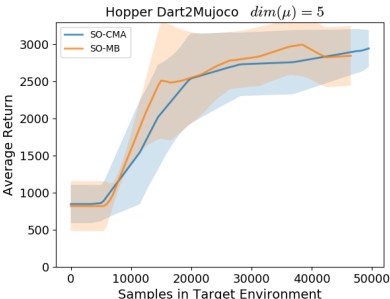

Figure 9: Comparison of SO-CMA and SO-MB for Hopper DART-to-MuJoCo transfer.

Model-based method provides more flexibility over CMA-ES and Bayesian optimization. For example, if the target environment changes over time, it may be desired to have $\mu$ also be time-varying. However, this would lead to a high dimensional search space, which might require significantly more samples for CMA-ES or Bayesian Optimization to solve the problem. If we can learn an accurate enough model from the data, we can use it to generate synthetic data for solving the problem.

However, there are two major drawbacks for Model-based method. The first is that to learn the dynamics model, we need to have access to the full state of the robot, which can be challenging or troublesome in the real-world. In contrast, CMA-ES and Bayesian optimization only require the final return of a rollout. Second, the Model-based method is significantly slower to run than the other methods due to the frequent training of the LSTM network.

