# OpenReview forum: "Policy Transfer with Strategy Optimization"
_ICLR.cc/2019/Conference_

### Official Review · AnonReviewer1 · 2018-11-02
**Interesting work with promising evaluation. Good evaluation.**

**Rating:** 6
**Confidence:** 4

**Review:**

The authors propose a policy transfer scheme which in the source domain simultaneously learns a family of policies parameterised by dynamics parameters and then employs an optimisation framework to select appropriate dynamics parameters based on samples from the target domain. The approach is evaluated on a number of simulated transfer tasks (either transferring from DART to MuJoCo or by introducing deliberate model inaccuracies).

This is interesting work in the context of system identification for policy transfer with an elaborate experimental evaluation. The policy learning part seems largely similar to that employed by Yu et al. 2017 (as acknowledged by the authors). This makes the principal contribution, in the eyes of this reviewer, the optimisation step conducted based on rollouts in the target domain. While the notion of optimising over the space of dynamics parameters is intuitive the question arises whether this optimisation step makes for a substantive contribution over the original work. This point is not really addressed in the experimental evaluation as benchmarking is performed against a robust and an adaptive policy but not explicitly against the (arguably) most closely related work in Yu et al. It could be argued, of course, that Yu et al. essentially use adaptive policy generation but they do explicitly learn dynamics parameters based on recent history of actions and observations. An explicit comparison therefore seems appropriate (or alternatively a discussion of why it is not required).

Another point which would, in my view, add significant value is explicit discussion of the baseline performances observed in the various experiments. For example, in the hopper experiment (Sec 5.2) the authors state that the baseline methods were not able to adapt to the new environment. Real value could be derived here if the authors could elaborate on why this is the case. The same applies in Sec 5.3-5.6.

(I would add here, as an aside, that I thought the notion in Sec 5.6 of framing the learning of policies for handling deformable objects as a transfer task based on rigid objects to be a nice idea. And not one this reviewer has come across before - though this could merely be a reflection of limited familiarity with the literature).

The experimental evaluation seems thorough with the above caveat of a seemingly missing benchmark in Yu et al. I would also encourage the authors to add more detail in the experimental section in the main text specifically with regards to number of trials run to arrive at variances in the figures as well as what metric these shaded areas actually signify.

A minor point: the J in equ 1 seems (to me at least) undefined. I suspect that it signifies the expected cumulative reward and was meant to be introduced in Sec 3 where the J may have been dropped from the latex?

If the above points were addressed I think this would make a valuable and interesting contribution to the ICLR community. As it stands I believe it is marginally below the acceptance threshold.

[ADDENDUM: given the author feedback and addition of the benchmark experiments requested I have updated my score.]


Pros:
———
- interesting work
- accessible
- effective
- thorough evaluation (though potentially missing a key benchmark)

Cons:
———
- potentially missing a key benchmark (and therefore seems somewhat incremental)
- only limited insight offered by the authors in the discussion of the experimental results
- some more details needed with regards to the experimental setup

---

> ### Author Response · Authors · 2018-11-15
> **Response to Reviewer 1**
>
> We thank the reviewer for the thoughtful comments! We have revised the paper to address the reviewer’s concerns, as detailed below.
>
> 1. Comparison to Yu et al. 2017
> We have added the comparison to UPOSI for the hopper, walker and halfcheetah examples (Figure 2, 3, 4 and 5). In general UPOSI transfers better than Hist, as expected. Our proposed method was able to notably outperform UPOSI in the hopper and walker example, while the results for halfcheetah example are comparable.
>
> 2. Discussion about baselines performances.
> We have added a new section (Section 6) that discusses the performance of the baseline methods for each example. Please refer to the revised text for more details. The related text is copied here for easy access:
>
> “We hypothesize that the large variance in the performance of the baseline methods is due to their sensitivity to the type of task being tested. For example, if there exists a robust controller that works for a large range of different dynamic parameters mu in the task, such as a bipedal running motion in the Walker2d example, training a Robust policy may achieve good performance in transfer. However, when the optimal controller is more sensitive to mu, Robust policies may learn to use overly-conservative strategies, leading to sub-optimal performance (e.g. in HalfCheetah) or fail to perform the task (e.g. in Hopper). On the other hand, if the target environment is not significantly different from the training environments, UPOSI may achieve good performance, as in HalfCheetah. However, as the reality gap becomes larger, the system identification model in UPOSI may fail to produce good estimates and result in non-optimal actions. Furthermore, Hist did not achieve successful transfer in any of the examples, possibly due to two reasons: 1) it shares similar limitation to UPOSI when the reality gap is large and 2) it is in general more difficult to train Hist due to the larger input space, so that with a limited sample budget it is challenging to fine-tune Hist  effectively.
>
> We also note that although in some examples certain baseline method may achieve successful transfer, the fine-tuning process of these methods relies on having a dense reward signal. In practice, one may only have access to a sparse reward signal in the target environment, e.g. distance traveled before falling to the ground. Our method, using an evolutionary algorithm (CMA), naturally handles sparse rewards and thus the performance gap between our method (SO-CMA) and the baseline methods will likely be large if a sparse reward is used.“
>
> 3. Experimental setup.
> We ran each trial with 3 random seeds and report the mean and one standard deviation in the plots. We have modified the first paragraph of the experiments section to emphasize this.
>
> 4. J in eq 1 undefined.
> Thanks for spotting this! It was indeed due to a typo in the latex file that dropped J in section 3. This has been fixed in the revision.

---

### Official Review · AnonReviewer3 · 2018-11-02
**Novel approach for adapting domain randomization policy for transfer**

**Rating:** 7
**Confidence:** 4

**Review:**

This paper presents a novel approach for adapting a policy learned with domain randomization to the target domain. The parameters for domain randomization are explicitly used as input to the network learning the policy. When run in the target domain, CMA-ES is used to search over these domain parameters to find the ones that lead to the policy with the best returns in the target domain.

This approach is a novel one in the space of domain randomization and sim2real work. The results show that it improves over learning robust policies and over one version of doing an adaptive policy (feedforward network with history input). This approach could

The paper is well written, clearly explained, has clear results, and also explains and evaluates alternate design choices in the appendix.

Pros:
- Demonstrated transfer across simulated environments
- Outperforms basic robust and adaptive alternatives
- Straightforward approach
Cons:
- Requires explicit domain randomization parameters as input to network. This restricts it from applying to work where the simulator is learned rather than parameterized in this way.

---

> ### Author Response · Authors · 2018-11-15
> **Response to Reviewer 3**
>
> We thank the reviewer for the valuable feedback!
>
> We share the reviewer’s concern that requiring explicit randomization parameters as inputs to the policy can be limiting for some applications. It is an interesting and important future direction to investigate how we can lift this limitation. One possible way is to use the method proposed by Eysenbach et al. [1], where a diverse set of skills is learned by maximizing how well a discriminative model can distinguish between different policies. Another possibility is to use the method in the work by Chen et al [2], as pointed out by Reviewer 2. They learned a latent representation of the environment variations by optimizing a latent input to the policy during the training.
>
>
> [1] Eysenbach, Benjamin, et al. "Diversity is All You Need: Learning Skills without a Reward Function." arXiv preprint arXiv:1802.06070 (2018).
> [2] Chen, Tao, et al. "Hardware Conditioned Policies for Multi-Robot Transfer Learning." NIPS, 2018.

---

### Official Review · AnonReviewer2 · 2018-11-08
**Simple technique with few assumptions for policy transfer. Questions regarding performance and novelty.**

**Rating:** 7
**Confidence:** 4

**Review:**

This paper introduces a simple technique to transfer policies between domains by learning a policy that's parametrized by domain randomization parameters. During transfer CMA-ES is used to find the best parameters for the target domain.

Questions/remarks:
- If I understand correctly, a rollout of a policy during transfer (i.e. an episode) contains 2000 samples. Hence, 50000 samples in the target environment corresponds to 25 episodes. Is this correct? Does fine-tuning essentially consists of performing 25 rollouts in the target domain?
- It seems that for some tasks, there is almost no finetuning happening whereas SO-CMA still outperforms domain randomization (Robust) significantly? How can this be explained? For example, the quadruped task (Fig 6a)  has no improvement for the SO-CMA method, yet it is significantly better than the domain randomization result. It seems that during the first episodes of finetuning, domain randomization and SO-CMA should be nearly equivalent (since CMA-ES will be randomly picking parameters mu). A very similar situation can be seen in Fig 5a
- Following up on my previous question: fig 4a does show the expected behavior (domain randomization and SO-CMA starting around the same value). However, in this case your method does not outperform domain randomization. Any idea as to why this is the case?
- It's difficult to understand how good/bad the performance of the various methods are without an oracle for comparison (i.e. just run PPO in the target environment).
- It seems that the algorithm in this work is almost identical to Hardware Conditioned Policies for Multi-Robot (Tao Chen et al. NIPS 2018), specifically section 5.2 in that paper seems very similar. Please comment.

Minor remarks:
- fig 5.a y-axis starts at 500 instead of 0.
- The reward for halfcheetah seems low, but this might be due to the custom setup.

---

> ### Author Response · Authors · 2018-11-15
> **Response to Reviewer 2**
>
> We thank the reviewer for the insightful comments! Below we discuss the questions and comments by the reviewer. We have also revised the text to address the comments.
>
> 1. Rollout number during fine-tuning
> During the fine-tuning stage, the policies interact with the target environment for 50,000 steps (corresponding to the results in Figure 2, 4 (a), 5 (a) and 6). In the case of fine-tuning Robust, Hist and UPOSI, we run PPO with 2,000 samples at each iteration, resulting in 50 iterations of PPO.
>
> In terms of the length of each rollout or trajectory, it has a maximum of 1,000 steps while the actual rollouts might be shorter due to early terminations.
>
> In our experiments, the fine-tuning phase in general takes between 100-300 rollouts depending on the task. We have also revised the related text (Appendix B.4) to make this more clear.
>
> 2. SO-CMA sometimes perform well without fine-tuning
> The reviewer’s concern about the SO-CMA sometimes achieving good performance with only one iteration is well taken. Upon further investigation, we think this is partly due to that the initial sampling distribution for CMA is chosen to be a Gaussian with the center of the mu domain as mean and a stdev of 0.25 (we use a mu domain of length 1 in each dimension). For the quadruped example, it turns out that the optimal solution of mu is close to the center of the mu domain and thus even in the first iteration of CMA, it might draw a sample that performs well. To validate this, we re-ran SO-CMA for the quadruped and the halfcheetah with CMA initial distribution to be a Gaussian with its mean randomly sampled and stdev be 0.5. This results in a more reasonable performance curve (as shown in Figure 5(a) and 6(a)) where the initial guess of CMA is sub-optimal and through the iterative optimization process it finds better solutions.
>
> 3. Performance of Robust in walker2d example
> For the walker2d example, fine-tuning a robust policy indeed achieved comparable performance to SO-CMA. We hypothesize that this is because Robust was able to discover a robust bipedal running gait that works near-optimally for a large range of different dynamic parameters mu. However, when the optimal controller is more sensitive to mu, Robust policies may learn to use over-conservative strategies, leading to sub-optimal performance (e.g. in HalfCheetah) or fail to perform the task (e.g. in Hopper).
>
> We do note that the fine-tuning process of the baseline methods relies on having a dense reward signal. In practice, one may only have access to a sparse reward signal in the target environment. Our method, using CMA, naturally handles sparse rewards and thus the performance gap between our method and the baseline methods will likely to grow if a sparse reward is used.
>
> We have added a new section that discusses the performance of baseline methods (Section 6). We refer the reviewer to the revised text for more details.
>
> 4. Oracle in the target environment
> We have trained oracle agents for our examples and added to the results (as seen in Figure 2-5). We trained the oracles for hopper, walker2d and halfcheetah environment for 3 random seeds with 1 million samples using PPO as in [1]. For the quadruped robot, we trained the oracle for 5 million samples as in [2]. Our method is able to achieve comparable or even better performance than the oracle agents.
>
> 5. Comparison to Tao Chen et al.
> We thank the reviewer for pointing out the work by Tao Chen et al. [3], which we missed during literature search. It is very interesting and highly relevant to ours. The most relevant part of the algorithm by Tao Chen et al is the HCP-I policy, where a latent variable representing the variations is trained along with the neural net weights using reinforcement learning. During the transfer stage, HCP-I is fine-tuned in the target environment with another RL process.
>
> Our method differs from HCP-I in two aspects. First, our policy takes the dynamic parameters as input, while HCP-I learns a latent representation of them. Second, during the transfer of the policy, we search in the low-dimensional mu space using CMA, instead of fine-tuning the entire neural network. Learning a latent representation of the variations in the dynamics can be more flexible, while searching in the mu space is more sample efficient and allows sparse reward when methods like CMA is used. It is an interesting future direction to see whether HCP-I can overcome large dynamics discrepancies like the ones in our examples and if using CMA for identifying the latent variables in HCP-I can result in a more sample-efficient transfer algorithm.
>
> We have added the HCP-I related discussions in related works section and conclusion section.
>
> [1] Schulman, John, et al. "Proximal policy optimization algorithms.".
> [2] Tan, Jie, et al. “Sim-to-Real: Learning Agile Locomotion For Quadruped Robots.”
> [3] Chen, Tao, et al. "Hardware Conditioned Policies for Multi-Robot Transfer Learning." NIPS, 2018.

---

> > ### Comment · AnonReviewer2 · 2018-11-28
> > **Satisfying revision. Upping my rating to 6 or 7.**
> >
> > Thanks for your detailed reply and revision. I think this strengthens this paper and I'd happily kick my rating up a notch to a 6 or 7. I'm not sure if I can still change my official rating, but I'm assuming the meta-reviewer will review this.
> >
> > In summary, I like the simplicity of this paper. This approach seems to perform on par with or better than more complicated meta-learning setups and is worthy of publication (it could at least serve as a good benchmark).

---

### Author Response · Authors · 2018-11-15
**Summary of paper revisions**

We have revised the paper based on the reviewers' comments. The main changes to the initial paper are the following:

- Added comparison to Yu et al. 2017 in Experiments (Figure 2-5).
- Added comparison to oracle agents, which are agents trained directly in the target environments (Figure 2-6).
- Re-ran SO-CMA for the single target example of half cheetah and quadruped to account for the initialization bias in the CMA-ES experiments (Figure 5, 6).
- Added a discussion section for a more detailed discussion of the results from different methods.
- Revised Related Work and Conclusion sections to include the work of Tao Cheng et al. 2018
- Fixed typos and figure inconsistencies as pointed out by the reviewers.

---

### Comment · Area_Chair1 · 2018-11-19
**responses and revised opinions & scores, based on author's replies?**

The detailed reviews are appreciated, as are the author's detailed replies.
As a next step, could the reviewers please advise as to whether the replies have influenced your evaluation
and your score for the paper?  Thank you in advance!
Note: to see the revision differences, select "Show Revisions" on the review page, and then select the check-boxes for the two versions you wish to compare.

-- area chair

---

> ### Comment · AnonReviewer3 · 2018-11-27
> **Revisions**
>
> The revisions make the paper quite a bit stronger and more complete. I'm maintaining my rating of 7-Accept.

---

### Meta-Review · Area_Chair1 · 2018-12-15
**Simple idea for sim2real fine tuning;  solid results;  with additional actual sim2real results, could be oral**

**Confidence:** 5
**Recommendation:** Accept (Poster)

**Metareview:**

The paper presents quite a simple idea to transfer a policy between domains by conditioning
the orginal learned policy on the physical parameter used in dynamics randomization.  CMA-ES then
finds the best parameters in the target domain. Importantly, it is shown to work well,
for examples where the dynamics randomization parameters do not span the parameters that are
actually changed, i.e., as is likely common in reality-gap problems.

A weakness is the size of the contribution beyond UPOSI (Yu et al. 2017), the closest work.
The authors now explicitly benchmark against this, with (generally) positive results.
AC: It would be ideal to see that the method does truly help span the reality gap, by seeing working sim2real transfer.

Overall, the reviewers and AC are in agreement that this is a good idea that is likely to have impact.
Its fundamental simplicity means that it can also readily be used as a benchmark in future sim2real work.
The AC recommend it be considered for oral presentation based on its simplicity, the importance of
the sim2real problem, and particularly if it can be demonstrated to work well on actual
sim2real transfer tasks (not yet shown in the current results).